Micropropagation of pokeweed (Phytolacca americana L.) and comparison of phenolic, flavonoid content, and antioxidant activity between pokeweed callus and other parts

Trunjaruen Attachai 1
Luecha Prathan 2
Taratima Worasitikulya worasitikulya@gmail.com 1
1 Salt-Tolerant Rice Research Group, Department of Biology, Faculty of Science, Khon Kaen University , Khon Kaen , Thailand
2 Department of Pharmacognosy and Toxicology, Faculty of Pharmaceutical Science, Khon Kaen University , Khon Kaen , Thailand
Winkler Robert
Electronic publication date: 2022 Feb 7
Publication date: 2022
Volume: 10
Electronic Location ID: e12892
Received 2021 Sep 20; Accepted 2022 Jan 16
Copyright: ©2022 Trunjaruen et al.
Copyright year: 2022
Copyright holder: Trunjaruen et al.
License: This is an open access article distributed under the terms of the Creative Commons Attribution License, which permits unrestricted use, distribution, reproduction and adaptation in any medium and for any purpose provided that it is properly attributed. For attribution, the original author(s), title, publication source (PeerJ) and either DOI or URL of the article must be cited.
License URL: https://creativecommons.org/licenses/by/4.0/

Keywords: Antioxidant, Callus, Chemical constituents, Micropropagation, Pokeweed

Funding: The Science Achievement Scholarship of Thailand (SAST) This study was financially supported by the Science Achievement Scholarship of Thailand (SAST). The funders had no role in study design, data collection and analysis, decision to publish, or preparation of the manuscript.

==============================
Background

Pokeweed (Phytolacca americana L.) is regarded as an invasive plant in many parts of the world but possesses therapeutic characteristics used for antitumor and rheumatism treatment. This study investigated the effects of auxins and four explants on pokeweed callus induction. The effects of cytokinins and combinations between cytokinins and NAA on shoot and root induction were also studied. TPC, TFC and antioxidant activity of calli were screened and compared with other pokeweed plant parts.

Methods

Four explants were used to induce callus using 2,4-D and IBA at 1, 2, 3 and 4 mg/l for each auxin. Direct shoot organogenesis from nodal explants was investigated using BAP, kinetin and TDZ (1, 2 and 4 mg/l for each cytokinin). Combined effects between cytokinins and NAA at 0.1, 0.2 and 0.3 mg/l were further simultaneously estimated with root induction. Calli derived from the leaves were compared with other plant parts for TPC, TFC and antioxidant activity using the Folin-Ciocalteu, AlCl3 colorimetric assay and DPPH assays, respectively.

Results

Results showed that MS medium containing 2 mg/l 2,4-D induced callus formation on leaf explants that provided highest fresh and dry weights. Three types of synthetic cytokinins as kinetin, TDZ and BAP were used for direct shoot organogenesis from pokeweed nodes. MS medium containing 2 mg/l kinetin was effective in stimulating normal shoots, with the largest number of shoots and leaves and the longest shoots. The combination between cytokinins and NAA showed no positive effect on shoot and root induction from pokeweed nodal explants. For TPC and TFC determination, pokeweed seeds and leaves possessed the highest phenolic and flavonoid contents, respectively. Highest phenolic content of pokeweed seeds led to lowest IC50 by DPPH assay. Phenolic content was higher than flavonoid content.

Conclusion

Results suggested promising conditions for callus induction. Leaf explants cultured on MS medium with 2 mg/l 2,4-D and nodal explants cultured on MS medium with 2 mg/l kinetin provided the largest number of normal shoots and leaves. NAA did not show positive effects on shoot and root induction when combined with cytokinins. Chemical constituent screening indicated that seeds and leaves provided highest TPC and TFC, respectively, while pokeweed calli contained higher phenolic than flavonoid content. This is the first report describing chemical constituent screening and antioxidant activity of calli and other parts of the pokeweed plant. Results provided significant information to further enhance bioactive compound contents of pokeweed calli using elicitation methods.

Introduction

Phytolacca americana L. or pokeweed (family Phytolaccaceae), is widely distributed as a native weed in North America and regarded as an invasive plant in many parts of the world (Balogh & Juhasz, 2008). Pokeweed seeds are phenolic-rich and possess interesting biological properties. Seed derivative americanin A shows antitumor activity for human colon cancer, while isoamericanin B and C inhibit tyrosinase activity (Jung et al., 2015; Petrillo et al., 2019). However, copious quantities of roots and seeds would be required for commercial purposes and plant tissue culture is now an effective technique to produce large numbers of plants and also metabolites that can be used for medicinal purposes.

Mass plant production within a limited time is the major advantage of the plant tissue culture technique (Bhoite & Palshikar, 2014). Callus can be induced from all parts of plants and used as materials for plant regeneration to stimulate production of bioactive compounds. Direct shoot organogenesis is a process whereby new shoots are directly produced from other vegetative parts (Bhatia & Bera, 2015). Callus induction and direct shoot organogenesis are different manipulation methods, and factors influencing both processes represent the balance between auxins and cytokinins. High ratios of auxins and cytokinins induce roots, while low ratios induce shoots from explants and equal ratios promote callus induction (Ikeuchi, Sugimoto & Iwase, 2013). However, optimal concentrations of auxins and cytokinins need to be estimated for plant species and specific types of explants. Therefore, it is necessary to select types of explants that are appropriate for micropropagation.

Most previous research about pokeweed micropropagation focused on direct shoot organogenesis (Zou et al., 2008 and El-Minisy et al., 2017) but research concerning callus induction of pokeweed is limited. Pokeweed callus induction was first established for cell suspension production. Sakuta, Hirano & Komamine (1991) induced calli from pokeweed stems and produced pokeweed cell suspension from the calli. Some phytochemicals, like betalains, can be produced from calli and cell suspension. Consequently, this study determined the effects of auxins and explant types on pokeweed callus induction to obtain calli that could be used for chemical constituents and antioxidant activity determination. The effects of PGRs and explants on direct shoot organogenesis were also evaluated for more efficient pokeweed micropropagation. Total phenolic and flavonoid contents and total antioxidant activity by DPPH assay were also evaluated to compare phytochemical contents between the callus and other parts of wild plants.

Materials & Methods

Plant materials

Pokeweed plants were cultivated, and seeds were obtained from ripened pokeweed fruits grown in Sakon Nakhon Province, Thailand (17°27′24.1″N 103°27′20.2″E). The plant was identified by Dr. Sakuntala Ninkaew, and the specimens were deposited at KKU herbarium (KKU No. 26576). For in vitro germination, pokeweed seeds were soaked in concentrated sulfuric acid for 10 min and then rinsed with running tap water. Seeds were then sterilized by agitation with 2% (v/v) sodium hypochlorite for 30 min and rinsed with sterilized distilled water for 5 min three times. The sterilized seeds were cultured on solid Murashige and Skoog (MS; 1962) medium supplemented with 30 g/l sucrose (pH 5.8). Cultures were incubated at 25 ± 2 °C with a 16/8 h (light/dark) cycle at 40 µmol m−2 s−1 light intensity. Media and equipment were sterilized by autoclaving at 121 °C for 20 min. Cotyledons, leaves, nodes and internodes were obtained from in vitro grown plants aged 2 months after germination. Cotyledons and leaf explants were cut into 1 × 1 cm squares, and 1 cm long nodal and internodal explants were used for callus induction and direct shoot organogenesis.

Callus induction

Cotyledons, leaves, nodes and internodes from in vitro grown pokeweed were used as initial explants. Solid MS medium fortified with 30 mg/l of sucrose (pH 5.8) was used as basal medium and supplemented with 2,4-D and IBA at 1, 2, 3 and 4 mg/l for both PGRs. MS medium without PGRs was considered as the control. Callus induction cultures were maintained under aseptic conditions at 25 ± 2 °C with a 16/8 h (light/dark) cycle at 40 µmol m−2 s−1 light intensity for four weeks before data collection.

Survival, response and callus formation percentages were calculated at week four after callus induction. Explants showing viable characteristics were considered as surviving. Response percentages were estimated from the surviving explants as changes in growth. Explants with callus formation were used to calculate callus induction percentage, fresh weight and dry weight. For dry weight, the calli were dried at 45 °C for 3 days.

Direct shoot organogenesis

Explants of cotyledons, leaves, nodes and internodes were used as plant materials. Solid MS medium with 30 g/l sucrose (pH 5.8) was supplemented respectively with three types of synthetic cytokinins as BAP, kinetin and TDZ applied for each cytokinin at 1, 2 and 4 mg/l. MS medium without PGRs was considered as the control. The cultures were maintained under aseptic conditions at 25 ± 2 °C with a 16/8 h (light/dark) cycle at 40 µmol m−2 s−1 light intensity for six weeks before data collection.

Survival percentages, response, callus formation, and root and shoot formation percentages were collected six weeks after culture for direct shoot organogenesis. Explants with shoots and roots were used to calculate shoot and root formation percentage. Number of shoots per explant, shoot length and number of leaves per shoot were evaluated simultaneously.

Combined effects of cytokinins and auxins

From four explant types, only nodal explants showed potential for shoot organogenesis. Therefore, nodes from in vitro grown pokeweeds were used as plant materials in this experiment. Solid MS media (pH 5.8) with three cytokinin conditions as 1 mg/l BAP, 2 mg/l kinetin and 1 mg/l TDZ were combined with four concentrations of NAA (0, 1, 2 and 3 mg/l). MS medium without PGRs was considered as the control. The cultures were maintained under aseptic conditions at 25 ± 2 °C with a 16/8 h (light/dark) cycle at 40 µmol m−2 s−1 light intensity for six weeks before data collection.

Percentages of survival, response, callus formation and root and shoot formation were collected at six weeks after culture for direct shoot organogenesis. Explants with shoots and roots were used to calculate shoot and root formation percentages. Number of shoots per explant, shoot length and number of leaves per shoot were evaluated simultaneously.

Chemical constituents and antioxidant activity

Calli induced from leaf explants cultured on MS medium added with 2 mg/l 2,4-D, leaves, roots and seeds were dried at 45 °C for 3 days. All samples were ground and then extracted with methanol. Briefly, 100 mg of powdered sample were extracted with 1 ml of methanol and sonicated for 30 min at 30 °C. The extraction was repeated three times and the extract solutions were assembled. Methanol was evaporated using a vacuum oven at 45 °C for 48 h. The extracts were used to determine chemical constituents and antioxidant activities.

Total phenolic content of the pokeweed callus extract was determined and compared with TPC of other plant parts using the Folin-Ciocalteu assay with some modifications following Mwamatope et al. (2020). Extracts at 20 µl were mixed with 100 µl of 10-fold diluted Folin-Ciocalteu solution and then added with 80 µl sodium carbonate (Na2CO3) in microplates. The microplates were incubated in darkness for 30 min. TPC was determined at 760 nm and calculated as mg gallic acid equivalent (GAE)/g dry weight extract.

Total flavonoid content of pokeweed callus extract was determined and compared with TFC of other plant parts based on the complex between flavonoid compounds and aluminum chloride (AlCl3) (Pekal & Pyrzynska, 2014). Extracts at 100 µl were mixed with 50 µl of 2% AlCl3 and 50 µl of water in microplates. The reaction was carried out in the dark for 30 min, determined at 425 nm and calculated as mg quercetin equivalent (QE)/g dry weight extract.

DPPH assay was carried out following the method of Mwamatope et al. (2020). The solution was incubated with DPPH reagent for 30 min at room temperature under dark condition. After incubation, absorbance was measured at 517 nm using a spectrophotometer. Methanol and ascorbic acid were used as negative and positive controls, respectively. Free radical scavenging activity was calculated by the following equation.

% free radical scavenging activity by DPPH = [1-(A sample / A control)] x 100

A sample means the absorbance of sample solution, while A control denotes the absorbance of negative control, with both measured at 517 nm.

Statistical analysis

All experiments were carried out with ten replications per treatment. Data were analyzed by one-way Analysis of Variance (ANOVA) and means were separated using Least Significant Difference (LSD; p < 0.05). Data were expressed as mean ± standard error of mean (SE). Statistical analysis was performed with Statistix 10 software.

Results

Effects of explant types and auxins on callus induction

Results showed that all types of explants survived and responded in all treatments without significant difference. Cotyledons and leaves only generated calli in media with 2,4-D, while roots were induced from explants by media supplemented respectively with IBA (Fig. 1A). Similar responses were also found in treatments of nodal and internodal explants cultured in the media with IBA (Fig. 1B). Interestingly, shoot formation was observed only in the treatments of nodal explants cultured on MS medium without PGRs and media with IBA, singly (Fig. 1C). This phenomenon showed the shoot organogenesis potential of pokeweed nodal explants.

Figure 1 In vitro calli and plantlets grown on MS media supplemented with different PGRs.

(A) Leaf explants cultured on MS medium with 4 mg/l IBA; (B) nodal explants cultured on MS medium with 3 mg/l IBA; (C) nodal explants cultured on MS medium at four weeks; (D) Callus with purple pigment inside the cells (arrow) and elongated clear cells (arrowhead); (E) leaf explants cultured on MS medium with 2 mg/l 2,4-D; (F) nodal explants cultured on MS medium at six weeks; (G) nodal explants cultured on MS medium with 1 mg/l BAP; (H) nodal explants cultured on MS medium with 1 mg/l TDZ; (I) nodal explants cultured on MS medium with 2 mg/l kinetin; scale = 1 cm (A–C, E–I); scale = 500 µm (D).

This experiment identified conditions that were suitable for callus induction of pokeweed. Calli from all treatments showed similar morphology as friable with pink to purple color. Observation under a stereomicroscope revealed elongated parenchymal cells, mostly with accumulated green or purple pigment and some elongated and clear without pigment (Fig. 1D). Considering callus induction percentage, all explant types cultured on media containing 2,4-D produced calli leading to 100% callus induction percentage in those treatments (P-value = 0.00; Table 1). Callus production was also observed in treatments of internodes and nodes cultured on media with IBA, ranging 37.50–62.50% and 56.25–81.25%, respectively. However, callus percentages from IBA treatments were significantly lower than those from 2,4-D treatments (P-value = 0.00; Table 1). Significantly highest callus fresh weight was obtained from nodes cultured on medium containing 3 mg/l IBA (Fig. 1B) and leaves cultured on medium containing 2 mg/l 2,4-D (Fig. 1E) at 0.81 g and 0.80 g, respectively while these two treatments also provided the highest dry weights of calli at 0.05 g (P-value = 0.00; Table 1). Callus induction percentage of leaf explants cultured on 2 mg/l 2,4-D reached 100%, while nodes cultured on 3 mg/l IBA provided only 62.50% (Table 1). Fresh and dry weights of calli generated from leaf explants (0.69–0.80 g and 0.04–0.05 g, respectively) were higher than calli from other explants (Table 1). Therefore, culture of leaf explants on MS medium supplemented with 2 mg/l 2,4-D was the most appropriate for callus induction from pokeweed.

Table 1 Callus induction percentage of different pokeweed explants affected by various types and concentrations of auxins.

Explants	PGRs	Callus induction (%)	Fresh weight (mg)	Dry weight (mg)	
	Auxins	Conc. (mg/l)				
Cotyledon	control	0	0.00 ± 0.0e	0.00 ± 0.0k	0.00 ± 0.0f	
2,4-D	1	100.00 ± 0.0a	0.44 ± 0.0g	0.03 ± 0.0c	
2	100.00 ± 0.0a	0.54 ± 0.0ef	0.03 ± 0.0c	
3	100.00 ± 0.0a	0.49 ± 0.0fg	0.03 ± 0.0c	
4	100.00 ± 0.0a	0.53 ± 0.0ef	0.03 ± 0.0c	
IBA	1	0.00 ± 0.0e	0.00 ± 0.0k	0.00 ± 0.0f	
2	0.00 ± 0.0e	0.00 ± 0.0k	0.00 ± 0.0f	
3	0.00 ± 0.0e	0.00 ± 0.0k	0.00 ± 0.0f	
4	0.00 ± 0.0e	0.00 ± 0.0k	0.00 ± 0.0f	
Leaf	control	0	0.00 ± 0.0e	0.00 ± 0.0k	0.00 ± 0.0f	
2,4-D	1	100.00 ± 0.0a	0.70 ± 0.0b	0.04 ± 0.0b	
2	100.00 ± 0.0a	0.80 ± 0.0a	0.05 ± 0.0a	
3	100.00 ± 0.0a	0.69 ± 0.1b	0.04 ± 0.0b	
4	100.00 ± 0.0a	0.75 ± 0.0ab	0.04 ± 0.0b	
IBA	1	0.00 ± 0.0e	0.00 ± 0.0k	0.00 ± 0.0f	
2	0.00 ± 0.0e	0.00 ± 0.0k	0.00 ± 0.0f	
3	0.00 ± 0.0e	0.00 ± 0.0k	0.00 ± 0.0f	
4	0.00 ± 0.0e	0.00 ± 0.0k	0.00 ± 0.0f	
Internode	control	0	0.00 ± 0.0e	0.00 ± 0.0k	0.00 ± 0.0f	
2,4-D	1	100.00 ± 0.0a	0.22 ± 0.0j	0.01 ± 0.0e	
2	100.00 ± 0.0a	0.21 ± 0.0j	0.01 ± 0.0e	
3	100.00 ± 0.0a	0.24 ± 0.0j	0.01 ± 0.0e	
4	100.00 ± 0.0a	0.22 ± 0.0j	0.01 ± 0.0e	
IBA	1	56.25 ± 14.7c	0.30 ± 0.0i	0.02 ± 0.0d	
2	62.50 ± 12.5c	0.35 ± 0.0hi	0.03 ± 0.0c	
3	56.25 ± 11.3c	0.35 ± 0.1hi	0.03 ± 0.0c	
4	37.50 ± 12.5d	0.38 ± 0.0h	0.03 ± 0.0c	
Node	control	0	0.00 ± 0.0e	0.00 ± 0.0k	0.00 ± 0.0i	
2,4-D	1	100.00 ± 0.0a	0.61 ± 0.0cd	0.04 ± 0.0b	
2	100.00 ± 0.0a	0.56 ± 0.0d−f	0.03 ± 0.0c	
3	100.00 ± 0.0a	0.56 ± 0.0c−e	0.03 ± 0.0c	
4	100.00 ± 0.0a	0.58 ± 0.0c−e	0.04 ± 0.0b	
IBA	1	62.50 ± 8.1c	0.62 ± 0.0c	0.04 ± 0.0b	
2	81.25 ± 9.1b	0.61 ± 0.0cd	0.04 ± 0.0b	
3	62.50 ± 8.1c	0.81 ± 0.0a	0.05 ± 0.0a	
4	56.25 ± 6.2c	0.59 ± 0.0c−e	0.04 ± 0.0b	
Notes.

Means ± SE followed by different letters are significantly different by ANOVA and least significant difference tests (p < 0.05). Different letters (a, b, c…) represent significant differences in columns (p < 0.05).

Effects of explant types and cytokinins on direct shoot organogenesis

Different types and concentrations of cytokinins were tested with cotyledon, leaf, internodal and nodal explants of pokeweed to induce direct shoot organogenesis. Six weeks after culture, percentages of root formation, shoot formation, number of shoots per explant, shoot length and number of leaves per shoot were recorded. Most treatments produced callus, while only nodal explants produced roots in the control treatment (Fig. 1F). Cotyledon explants turned from green to purple in treatments of control and kinetin without any other changes, while color change was also found when leaf explants were cultured on the control medium.

Shoot formation percentages were calculated to identify suitable conditions for direct shoot organogenesis. Results showed that only nodal explants produced novel shoots that emerged directly from the leaf axillary. All treatments provided 100% shoot formation percentage, except for the control that gave 37.50% and significantly lower than the others (P-value = 0.00; Table 2). All treatments provided new shoots from the explants, but some abnormalities were observed from these shoots and leaves. In the control treatment, generated shoots gave expanding leaves with slender stems, which were considered normal characteristics (Fig. 1F). Shoots from BAP and TDZ treatments showed aberrations such as curling at the leaf margin and fleshy thick stems (Figs. 1G and 1H), whereas more slender and less fleshy stems with normal leaves were observed in treatments of kinetin (Fig. 1I). The number of shoots per explant, shoot length and number of leaves per shoot were also recorded. Treatment of 1 mg/l TDZ provided significantly largest number of shoots per explant (2.43 shoots), while nodal explants cultured on medium with 2 mg/l kinetin gave significantly longest shoots and largest number of leaves per shoot as 5.04 cm and 10.93 leaves, respectively (P-value = 0.00; Table 2). Interestingly, all treatments except for the control generated callus at the basal part of nodal explants. In each type of explant, cytokinin concentrations at 1, 2 and 1 mg/l gave significantly highest number of shoots, shoot length and number of leaves for BAP, kinetin and TDZ, respectively. Therefore, these concentrations were used in the basal medium to investigate the combined effects of cytokinins and auxins on direct shoot organogenesis.

Table 2 Direct shoot organogenesis from nodal explants of pokeweed influenced by various types and concentrations of cytokinins.

Cytokinins conc. (mg/l)	Root formation (%)	Shoot formation (%)	Number of shoots	Shoot length (cm)	Number of leaves	
Control	
0	31.25 ± 9.1a	37.50 ± 8.1b	1.00 ± 0.0c	4.41 ± 0.4ab	6.06 ± 0.4d	
BAP	
1	0.00 ± 0.0b	100.00 ± 0.0a	2.18 ± 0.1ab	4.18 ± 0.1b	9.50 ± 0.3ab	
2	0.00 ± 0.0b	100.00 ± 0.0a	2.06 ± 0.1ab	3.82 ± 0.1bc	8.81 ± 0.5bc	
4	0.00 ± 0.0b	100.00 ± 0.0a	1.25 ± 0.1c	2.49 ± 0.1e	5.56 ± 0.6d	
Kinetin	
1	0.00 ± 0.0b	100.00 ± 0.0a	1.25 ± 0.1c	4.93 ± 0.2a	10.37 ± 0.5ab	
2	0.00 ± 0.0b	100.00 ± 0.0a	1.81 ± 0.1b	5.04 ± 0.2a	10.93 ± 0.7a	
4	0.00 ± 0.0b	100.00 ± 0.0a	2.12 ± 0.2ab	3.82 ± 0.1bc	9.50 ± 0.8ab	
TDZ	
1	0.00 ± 0.0b	100.00 ± 0.0a	2.43 ± 0.2a	3.12 ± 0.2de	7.18 ± 0.5cd	
2	0.00 ± 0.0b	100.00 ± 0.0a	2.18 ± 0.1ab	2.93 ± 0.1de	7.06 ± 0.6d	
4	0.00 ± 0.0b	100.00 ± 0.0a	1.81 ± 0.1b	3.44 ± 0.2cd	6.93 ± 0.5d	
Notes.

Means ± SE followed by different letters are significantly different by ANOVA and least significant difference tests (p < 0.05). Different letters (a, b, c…) represent significant differences in columns (p < 0.05).

Combined effects of cytokinins and auxins on direct shoot organogenesis

According to the previous study, 1 mg/l BAP, 2 mg/l kinetin and 1 mg/l TDZ as suitable conditions for direct shoot organogenesis were combined with 0.1, 0.2 and 0.3 mg/l NAA to optimize conditions for direct shoot organogenesis and root induction simultaneously. The same parameters with the previous experiments were collected after culture for six weeks. Results showed that survival and response percentages were not affected by these combinations, leading to 100% in all treatments. However, shoots from the control have roots emerged from the base, while the media containing both cytokinin and NAA could not produce root from the explants (Table 3).

Table 3 Combined effects of cytokinins and NAA on direct shoot organogenesis from pokeweed nodal explants.

PGRs						
Cytokinins	NAA conc. (mg/l)	Roots formation (%)	Shoots formation (%)	Shoots number	Shoot length	Leaves number	
Control		31.25 ± 9.15a	37.50 ± 8.18b	1.00 ± 0.00e	4.41 ± 0.49ab	6.06 ± 0.49cd	
1 mg/l BAP	0	0.00 ± 0.00b	100.00 ± 0.00a	2.18 ± 0.16a−c	4.18 ± 0.13b	9.50 ± 0.37b	
	0.1	0.00 ± 0.00b	100.00 ± 0.00a	2.06 ± 0.19a−d	3.86 ± 0.11b	7.25 ± 0.50c	
	0.2	0.00 ± 0.00b	100.00 ± 0.00a	1.68 ± 0.17cd	3.80 ± 0.11b	6.43 ± 0.34cd	
	0.3	0.00 ± 0.00b	100.00 ± 0.00a	1.43 ± 0.15de	3.18 ± 0.18cd	7.12 ± 0.40c	
2 mg/l KIN	0	0.00 ± 0.00b	100.00 ± 0.00a	1.81 ± 0.16b−d	5.04 ± 0.22a	10.93 ± 0.70a	
	0.1	0.00 ± 0.00b	100.00 ± 0.00a	2.50 ± 0.32a	3.16 ± 0.19cd	5.25 ± 0.44d	
	0.2	0.00 ± 0.00b	100.00 ± 0.00a	2.50 ± 0.30a	2.98 ± 0.25d	5.25 ± 0.49d	
	0.3	0.00 ± 0.00b	100.00 ± 0.00a	1.93 ± 0.21a−d	2.79 ± 0.23d	6.81 ± 0.66c	
1 mg/l TDZ	0	0.00 ± 0.00b	100.00 ± 0.00a	2.43 ± 0.25ab	3.12 ± 0.21d	7.18 ± 0.57c	
	0.1	0.00 ± 0.00b	100.00 ± 0.00a	2.37 ± 0.28ab	3.21 ± 0.18cd	7.31 ± 0.29c	
	0.2	0.00 ± 0.00b	100.00 ± 0.00a	1.73 ± 0.26cd	2.60 ± 0.15d	6.00 ± 0.34cd	
	0.3	0.00 ± 0.00b	100.00 ± 0.00a	1.68 ± 0.23cd	2.96 ± 0.18d	6.62 ± 0.42c	
Notes.

Means ± SE followed by different letters are significantly different according to ANOVA and Least Significant Different Test (p < 0.05). Different letters (a, b, c…) represent significant differences in columns (p < 0.05).

The control treatment gave 37.50% shoot formation, significantly lower than percentages from other treatments that all recorded 100% (P-value = 0.00; Table 3). Abnormal morphology increased when NAA concentration increased, for example curling at the leaf margin, non-expanding leaves and fleshy stems. Media supplemented with 2 mg/l kinetin combined with 0.1 and 0.2 mg/l NAA showed significantly largest number of shoots per explant (2.5 shoots; P-value = 0.00). For shoot length and number of leaves per shoot, the highest parameters were obtained from single treatment of 2 mg/l kinetin as 5.04 cm and 10.93 leaves, respectively (P-value = 0.00; Table 3). Interestingly, these parameters tended to decrease when concentration of NAA combined with each type of cytokinin increased (Table 3). Therefore, parameters from media supplemented with only each cytokinin were higher than those from treatments of cytokinin-auxin combination.

Chemical constituents and antioxidant activity

Phenolic and flavonoid contents and IC50 values for DPPH assay of the four explants were determined as preliminary screening of some bioactive pokeweed compounds. Parameters of calli, leaves, roots and seeds of pokeweed were compared (Table 4). Results demonstrated that pokeweed seeds provided the highest phenolic content as 155.83 mg GAE/g dry weight extract, while calli, leaves and roots had significantly lower phenolic content than seeds ranging from 14.03 to 26.03 (P-value = 0.00). The flavonoid content of pokeweed leaves was significantly higher than other explants (57.31 mg QE/g dry weight extract). Pokeweed seeds and roots contained lower flavonoid content than leaves as 49.81 and 33.46 mg QE/g dry weight extract, respectively while pokeweed calli had the lowest flavonoid content (11.73 mg QE/g dry weight extract; P-value = 0.00).

Table 4 TPC, TFC and IC50 by DPPH assay of different types of pokeweed explants.

Explant
types	TPC
(mg GAE/g extract)	TFC
(mg QE/g extract)	IC50 by DPPH assay
(µg/ml)	
Callus	26.03 ± 0.6b	11.73 ± 0.5d	341.99 ± 15.3b	
Leaf	28.56 ± 0.4b	57.31 ± 0.7a	228.70 ± 4.0c	
Root	14.03 ± 0.3b	33.46 ± 1.2c	1125.51 ± 21.5a	
Seed	155.83 ± 8.6a	49.81 ± 1.3b	12.66 ± 0.6d	
Ascorbic acid	–	–	3.91 ± 0.0d	
Notes.

Means ± SE followed by different letters are significantly different by ANOVA and least significant different tests (p < 0.05). Different letters (a, b, c…) represent significant difference in columns (p <  0.05).

Antioxidant activities of the four pokeweed explants were determined and compared with ascorbic acid as control using DPPH assay and expressed inhibitory concentration at 50% (IC50). Pokeweed seeds and ascorbic acid had significantly lowest IC50 at 12.66 and 3.91 µg/ml, respectively. Higher IC50 values were obtained from leaves and calli at 228.70 and 341.99 µg/ml, respectively, while roots showed the highest IC50 at 11256.51 µg/ml. These results demonstrated that pokeweed calli from in vitro culture generated phenolic and flavonoid compounds that were responsible for antioxidant activity determined by DPPH assay.

Discussion

Different responses of explants to auxins in pokeweed callus induction

Different types of pokeweed explants from aseptic plants were cultured on MS medium with 2,4-D and IBA at various concentrations to investigate the effects of explant types and auxins on callus induction efficiency. Survival and response percentage from all treatments were not affected but fresh and dry weight of calli were strongly influenced. Remarkably, callus from any part of pokeweed showed green to purple, resulting from the betacyanin pigments accumulated in callus cells (Sakuta, Hirano & Komamine, 1991).

Our results found that calli derived from leaf explants had larger fresh and dry weights than calli derived from other explants. The success of callus induction from leaf explants was also reported in Vanda sp. as callus induction from leaf segments (Budisantoso, Amalia & Kamsinah, 2017). Possible reasons for the success were high stomatal density and spongy mesophyll that facilitated nutrients and PGR uptake, therefore, leaf explants provided high callus efficiency (Zhang et al., 2020). These reasons were also valid for callus induction from cotyledons that have stomata and spongy mesophyll. Recent results showed that fresh and dry weight of cotyledon-derived calli were similar to calli from leaf explants. In lotus (Nelumbo nucifera), four explant types were tested for callus induction. Results showed that immature cotyledons were suitable for callus induction of lotus, with 20% callus formation after three weeks of culture (Deng et al., 2020). However, our results showed adverse results of nodal and internodal explants, while some previous studies demonstrated the positive effects of those explants. Kumlay & Ercisli (2015) recorded high performance for callus induction from nodal explants of potato (Solanum tuberosum). They explained that potato leaves have a large surface area with high rate of water loss, while lower surface area and stomata of nodal explants led to better efficiency of callus induction. Our results confirmed that differences in explant histology affected callus induction efficiency and response of explants to PGRs.

The balance of PGRs in growth media is a vital factor that affects plant callus induction. Equal ratios of auxins and cytokinins promoted callus induction (Ikeuchi, Sugimoto & Iwase, 2013) but appropriate types and concentrations of auxins and cytokinins must first be considered as PGRs for specific plant species. Most studies reported that exogenous auxins were necessary for callus induction. Therefore, here, we focused on the effects of auxin types and concentrations on callus induction of pokeweed. In this study, 2,4-D and IBA were applied with different pokeweed explants. Results showed that callus induction media containing 2,4-D induced calli from all types of explants, consistent with previous studies. Callus induction from bamboo (Dendrocalamus hamiltonii) was carried out by shoot tip culture on medium with 3 mg/l 2,4-D and 1 mg/l benzyladenine (Zang et al., 2016). Callus induction medium with 2 mg/l 2,4-D was identified as the optimal condition for landrace rice (Oryza sativa) from Thailand (Trunjaruen et al., 2020). Auxins, e.g., 2,4-D and NAA in callus induction medium were reported to promote expression of important genes like LBD family and ARF19 (Ikeuchi, Sugimoto & Iwase, 2013), resulting in callus formation. Supplementation of 2,4-D also induced several genes involving signal transmission and overexpression of transcription factors ZmBBM in maize (Zea mays) that induced callus formation from their scutella (Du et al., 2019). All these genes have been proved for involvement of auxins with callus formation under in vitro conditions.

Some studies have investigated pokeweed cell suspensions. Medium supplemented with 1 mg/l 2,4-D was used for callus induction from stem explants and cell suspension establishment from those calli (Sakuta, Takagi & Komamine, 1986; Sakuta, Hirano & Komamine, 1991). However, reports about the effects of explant types and other auxins on callus induction are limited. Therefore, this study determined improved and reliable conditions for pokeweed callus induction. The calli obtained from this study could be also extracted and determined phenolic, flavonoid content, and antioxidant activity.

Direct shoot organogenesis influenced by explant and cytokinin types

Callus induction is a vital step for plant micropropagation because callus can be applied as the initial materials for indirect organogenesis. However, our preliminary experiments demonstrated that indirect shoot organogenesis could not be induced from calli, therefore, direct shoot organogenesis was an alternative method to produce in vitro pokeweed plantlets. The plantlets can be used for pokeweed propagation, while leaves from the plantlets can also be used as explants for callus induction.

In a recent study, we investigated the effects of cytokinins on direct shoot organogenesis of pokeweed from different types of explants. Our results showed that new shoots were only generated from nodal explants and were induced in all treatments. Several studies reported the success of direct shoot organogenesis from nodal explants. Direct shoot organogenesis of maize was carried out with nodal explants cultured on medium supplemented with 1.8 mg/l benzyladenine (Mushke, Yarra & Bulle, 2016), while compared with three different nodal explants, Tylophora indica showed the highest efficiency of direct shoot organogenesis (Najar et al., 2018). New shoots emerged from the leaf axils of nodal explants, possibly influenced by apical dominance that promoted multiple shoots (Oliveira et al., 2013).

Our findings revealed that nodal explants from all treatments provided new shoots. The control treatment gave lowest shoot formation percentage, while other treatments reached 100% (Table 2). These results proved that applications of exogeneous cytokinins had positive effects on direct shoot organogenesis, consistent with many previous studies. A concentration of 3 mg/l BAP was reported as the optimal condition over other cytokinins for direct shoot regeneration of potato tree (Solanum erianthum; Sarkar & Banerjee, 2020). In direct shoot organogenesis of Cryptocoryne wendtii, shoot tips in the control treatment gave lower shooting percentage than treatments of BAP and kinetin, while the appropriate cytokinin conditions for direct shoot organogenesis was 3 mg/l kinetin (Klaocheed et al., 2020). For direct shoot organogenesis of apple, appropriate types and concentrations of cytokinins mainly depended on genotypes and initial explants (Magyar-Tabori et al., 2010). The AHK4/WOL cytokinin receptor is the major component in shoot regeneration and induces many downstream genes. Overexpression of ARR1 and WUS, which are normally upregulated by AHK4/WOL, promoted shoot regeneration in a cytokinin-free medium (Ikeuchi et al., 2019). This information proved that cytokinins promote shoot regeneration through the regulation of gene expression.

Results showed that 1 mg/l TDZ provided the largest number of shoots per explant, while shoots from treatments of TDZ showed abnormalities like thick and succulent stems and curling leaves. These characteristics were considered as hyperhydricity or vitrification, as phenomena that always occur in plants grown under in vitro conditions (Kevers et al., 2004). Micropropagated pear showed higher hyperhydricity in the shoots from TDZ treatments than benzyladenine and kinetin (Kadota & Niimi, 2003). Therefore, treatment of 2 mg/l kinetin was considered to be the most suitable conditions for pokeweed direct shoot organogenesis because novel shoots were not abnormal, with longest short length and highest number of leaves as 1.81 shoots per explant, 5.04 cm length and 10.94 leaves per shoot (Table 2).

Previous studies reported protocols for direct shoot organogenesis of pokeweed. MS medium with 2.0 mg/l benzyladenine and 0.2 NAA mg/l was appropriate for direct shoot induction from stem segments of pokeweed (Zou et al., 2008). MS medium supplemented with 2 mg/l BAP and 0.2 mg/l NAA provided 2.6 shoots generated from stem segments, higher than our results

(El-Minisy et al., 2017). However, micropropagated shoots reported by El-Minisy et al. (2017) were longer with greater numbers of leaves than previous studies. As well as direct shoot organogenesis, callus generation was induced from the base of micropropagated shoots in all treatments except for the control. This proved that the application of cytokinins stimulated callus formation through the cytokinin-mediated callus formation pathway (Ikeuchi, Sugimoto & Iwase, 2013).

Our results provided promising conditions for direct shoot organogenesis, which gave abundant pokeweed shoots that could be further propagated. The leaves from in vitro plantlets could also be used as explants for callus induction.

Combined effects between cytokinins and auxin

Three cytokinins as BAP, TDZ and kinetin at 1, 2 and 1 mg/l, respectively were selected from the highest efficiency of shoot organogenesis from the previous experiment. These treatments were combined with 0.1, 0.2 and 0.3 mg/l NAA. Results showed that kinetin treatments tended to increase the number of shoots, while other parameters decreased (Table 3). Application of NAA with cytokinins produced vitrified shoots with thick and fleshy stems and abnormal leaves.

Most previous studies reported positive effects of cytokinin-auxin combination on shoot organogenesis. Multiple shoots of Ruta graveolens were induced from shoot tip meristems by MS medium supplemented 2.25 mg/l benzyladenine and 0.47 mg/l NAA (Faisal et al., 2018). Positive effects of the combination were also reported in pokeweed shoot induction. Zou et al. (2008) induced new pokeweed shoots from stem segments using 2 mg/l BA and 0.2 mg/l NAA, while 1 mg/l BAP combined with 0.2 mg/l NAA was found to be suitable for shoot regeneration from stem cuttings (El-Minisy et al., 2017). However, our results showed that single cytokinins gave better results than cytokinin-auxin combination treatments. Similar results were also found in shoot multiplication of Amygdalus communis apical shoot tips. Shoot formation was not observed in treatments containing benzyladenine and kinetin combined with NAA (Akbaş et al., 2009). In this study, we hypothesized that the addition of NAA with cytokinins may induce shoots and roots, simultaneously. However, the results indicated that media fortified with cytokinins and NAA did not affect pokeweed root induction. A high ratio between cytokinins and auxins was suitable for shoot induction (Ikeuchi, Sugimoto & Iwase, 2013). We also found that micropropagated shoots showed some vitrified characters when NAA concentration increased. Hyperhydricity may result from excessive concentrations of PGRs (Karimpour et al., 2013). Therefore, we suggested that 2 mg/l kinetin without supplementation with NAA could induce multiple shoots of pokeweed from nodal explants.

Chemical constituents and antioxidant activity

Among all callus induction treatments, leaf explants cultured on MS medium added with 2 mg/l 2,4-D gave calli with the highest dry weight. Nazir et al. (2020) found that the highest biomass purple basil (Ocimum basilicum L. var purpurascens) calli provided the highest phenolic and flavonoid contents. Therefore, calli possessing the highest dry weight were used to determine phenolic and flavonoid contents and antioxidant activity and compared with other pokeweed plant parts.

Different parts of pokeweed possess diverse major secondary metabolites. Here, phenolic and flavonoid contents of various pokeweed explants were determined to preliminarily screen the chemical constituents, and the results showed that pokeweed seeds provided the highest phenolic content. Neo-lignans and phenolic compounds accumulated mainly in pokeweed seeds (Bailly, 2021). These phenolic compounds have many medicinal properties and may be responsible for high antioxidant activity, as shown by the lowest IC50 value in DPPH assay. Other explants showed lower phenolic content, especially in roots. Results proved that phenolic compounds were not the major accumulated compounds in leaves and roots. Leaves accumulated the highest flavonoid content compared with other explants. Flavonoid content was also higher than phenolic content in leaves and roots. The major secondary metabolites stored in leaves and roots were not phenolics and flavonoids. Research on chemical constituents of pokeweed leaves is limited but it has been reported that the leaves are green and their veins are always purple due to the accumulation of betacyanins (Jerz et al., 2008), while there was strong evidence that triterpenes and phytosterols accumulated in pokeweed roots (Choe et al., 2020; Jeong et al., 2004).

Previous studies of pokeweed callus chemical constituents focused on betacyanin extraction. Callus induced from leaf explants also showed purple pigments of betacyanins (Kobayashi et al., 1995; Fig. 1D), while betacyanins were also extracted from cell suspension maintained under aseptic conditions (Sakuta, Hirano & Komamine, 1991). Recent studies reported that low levels of phenolic and flavonoid contents accumulated in pokeweed calli resulted in low antioxidative potential. Similar results were found in most research. Calli of Helicteres angustifolia produced lower phenolic, flavonoid and saponin contents than the wild roots (Yang et al., 2019). Phenolic and flavonoid contents of blueberry (Vaccinium corymbosum) calli were lower than field-grown and in vitro leaves (Kolarevic et al., 2021). The phenomenon could be explained by the dedifferentiated property of the callus that resulted in lesser production and accumulation of complicated metabolites (Kolarevic et al., 2021; Yang et al., 2019). Lower metabolite contents in callus may result from limited metabolite production by in vitro explants but could be enhanced by callus elicitation to reduce cost and time to obtain pokeweed seeds and roots possessing major compounds of pokeweeds.

Conclusions

Pokeweed plants possess medicinal properties and may be applied in agricultural and medicinal fields. This study investigated the effects of explant types and auxins on pokeweed callus induction. Results showed that all types of explants generated callus by the application of exogenous 2,4-D. Suitable conditions for callus induction were culture of leaf explants on MS medium supplemented with 2 mg/l 2,4-D. This study proposed an improved and reliable system for pokeweed callus induction. In direct shoot organogenesis, pokeweed nodes were the only explant type for multiple shoot induction by cytokinins. Shoots with normal characteristics and largest number of shoots and leaves with the longest shoots were induced from MS medium fortified with 2 mg/l kinetin. We also found that inhibitory effects of combination with vitrified shoots increased when NAA concentration was increased. This study proposed an effective method for fast multiplication of pokeweed to utilize the plant in several fields. Pokeweed seeds showed the highest phenolic contents and lowest IC50 by DPPH assay, while leaves contained the highest flavonoid contents. Pokeweed calli showed higher phenolic than flavonoid content. This is the first report describing chemical constituent screening and antioxidant activity of calli and other parts of the pokeweed plant.

Supplemental Information

Supplemental Information 1 Raw data of callus induction of different pokeweed explants affected by various types and concentrations of auxins

Click here for additional data file.

Supplemental Information 2 Raw data of direct shoot organogenesis from nodal explants of pokeweeds influenced by various types and concentrations of cytokinins

Click here for additional data file.

Supplemental Information 3 Raw data Combined effects of cytokinins and NAA on direct shoot organogenesis from pokeweed nodal explants

Click here for additional data file.

Supplemental Information 4 Raw data of TPC, TFC, and IC50 by DPPH assay of different types of pokeweed explants

Click here for additional data file.

Supplemental Information 5 Growth performance of pokeweed plantlets regenerated from nodal segments using combination of various type of cytokinin and various concentrations of NAA

Combination effects of cytokinins and NAA on direct shoot organogenesis from pokeweed nodal explants.

Click here for additional data file.

Additional Information and Declarations

Competing Interests

Author Contributions

Data Availability

The authors declare there are no competing interests.

Attachai Trunjaruen conceived and designed the experiments, performed the experiments, analyzed the data, prepared figures and/or tables, authored or reviewed drafts of the paper, and approved the final draft.

Prathan Luecha conceived and designed the experiments, prepared figures and/or tables, authored or reviewed drafts of the paper, and approved the final draft.

Worasitikulya Taratima conceived and designed the experiments, analyzed the data, prepared figures and/or tables, authored or reviewed drafts of the paper, supervise this project, and approved the final draft.

The following information was supplied regarding data availability:

The data is available in the Supplementary Files.

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
