# Peer review of "Micropropagation of pokeweed (Phytolacca americana L.) and comparison of phenolic, flavonoid content, and antioxidant activity between pokeweed callus and other parts"

_PeerJ, doi:10.7717/peerj.12892_

## Round 0.1 · original submission · Major Revisions

Both reviewers give recommendations you should address in your revised manuscript. Please ensure in the improved version:

- The research question must be clearly stated. The introduction, question, results, discussion, and conclusion must be congruent.

- Please pass your manuscript to a professional proofreading service.

Reviewer 1 ·

Basic reporting

Title : In vitro propagation, chemical constituents and antioxidant activity from different explants of Phytolacca americana L.
Q1 : It is necessary to bring out the work aim which is the comparison between the phytochemical content at callus level and at other plant part level.
Abstract :
“Factors affecting callus induction and direct shoot organogenesis were investigated and compared with total phenolic content (TPC), total flavonoid content (TFC) and total antioxidant activity. Four different explants were cultured on MS medium containing various concentrations of 2,4-D and IBA”.
Q1: This paragraph does not reflect the adopted methodology
“This is the first report describing comparative micropropagation using chemical constituents and antioxidant activity from different explants of pokeweed. Results will provide significant information to further enhance bioactive compound contents of pokeweed calli using elicitation methods.
Q2: This paragraph should be reformulated. It is not clear.

Experimental design

Materials and methods:
Q1 : Line 78 : Rinsing has been carried out for how long?
Q2 : Line 88 : while describing the callus induction, the concentrations (1,2, 3 and 4 mg / l) are for which hormone, 2,4-D or IBA? It is not clear. Likewise, for line 100.
Q3 : Line 89 : What are the callus culture conditions (Photoperiod)?
Q4 : Line 105 : what is the callus origin used for the chemical constituents determination ? Why you did not perform the assays using the same explants used for callus induction.
Q5 : Line 95 et 105 : At what temperature the callus were dried ? 70 ° C or 45 ° C?
Q6 : Line 107 : At what temperature the sonication was performed ?

Validity of the findings

Results
Q1 : The direct organogenesis work is interesting, but these are only preliminary tests. Rooting tests could be done to value this work. Besides, it is quite normal that the rooting rates are zero since you only used cytokinins.
Q2: There is a lack of a connective link between the different parts of the work. The insertion of the direct organogenesis part is not done in a proper way.

Q3 : In order to achieve callogenesis induction, why you did not try the auxin and cytokinin combination?
Q4 : Line 127 : You argued in part MM that the antioxidant activity is calculated as a percentage : Free radical scavenging activity was calculated by the following equation : % free radical scavenging activity by DPPH = [1-(A sample / A control)] x 100. In table 3, the results are presented only in IC50 by DPPH assay (µg / ml).

Additional comments

Discussion :
Chemical constituents and antioxidant activity: This part needs to be further improved and amended with more recent references. The authors should place more emphasis on the work done on the extraction and assay of bioactive molecules from tissue cultures.

Reviewer 2 ·

Basic reporting

Refer attachment.

Experimental design

Some methods used need revision. Refer attachment.

Validity of the findings

Refer attachment.

Annotated reviews are not available for download in order to protect the identity of reviewers who chose to remain anonymous.

---

## Round 0.2 · accepted · Accept

Thanks for improving the writing and the methods (with statistics).

Reviewer 1 ·

Basic reporting

no comment

Experimental design

no comment

Validity of the findings

no comment

Additional comments

After reading the second version of the paper “ Micropropagation of pokeweed (Phytolacca americana L.) and comparison of phenolic, flavonoid content, and antioxidant activity between pokeweed callus and other parts '' I find that the authors have answered to all clarifications and have answered to all requested changes.
Thus, I believe that their study deserve to be published.